# Pre-Pregnancy Adherence to the Mediterranean Diet and Gestational Diabetes Mellitus: A Case-Control Study

**DOI:** 10.3390/nu11051003

**Published:** 2019-05-01

**Authors:** Rocío Olmedo-Requena, Julia Gómez-Fernández, Carmen Amezcua-Prieto, Juan Mozas-Moreno, Khalid S. Khan, José J. Jiménez-Moleón

**Affiliations:** 1Department of Preventive Medicine and Public Health, University of Granada, 18016 Granada, Spain; rocioolmedo@ugr.es (R.O.-R.); carmezcua@ugr.es (C.A.-P.); jjmoleon@ugr.es (J.J.J.-M.); 2Consortium for Biomedical Research in Epidemiology & Public Health (CIBER Epidemiología y Salud Pública-CIBERESP), 28029 Madrid, Spain; 3Instituto de Investigación Biosanitaria ibs.GRANADA, 18071 Granada, Spain; 4Obstetrics and Gynecology Service, Jaen Hospital Complex, 23007 Jaén, Spain; gomezfernandezjulia@gmail.com; 5Obstetrics and Gynecology Service, Virgen de las Nieves University Hospital, 18014 Granada, Spain; 6Departament of Obstetrics and Gynecology, University of Granada, 18016 Granada, Spain; 7Women’s Health Research Unit, Barts and the London School of Medicine, Queen Mary University London, London E1 4NS, UK; profkkhan@gmail.com; 8Multidisciplinary Evidence Synthesis Hub (mEsh), Barts and the London School of Medicine and Dentistry, Queen Mary University of London, London E1 4NS, UK

**Keywords:** Mediterranean diet (MD), gestational diabetes mellitus (GDM), pregnancy, maternal nutrition, lifestyles

## Abstract

Gestational diabetes mellitus (GDM), an important public health problem that affects mothers and offspring, is a common metabolic disorder. We evaluated the effect of the pre-pregnancy Mediterranean diet (MD) level of exposure on the odds of GDM development. A case-control study (291 GDM cases and 1175 controls without GDM) was conducted in pregnant women. Pre-pregnancy dietary intake was assessed using a validated food frequency questionnaire to calculate an MD adherence index (range score 0–9: low ≤ 2; middle 3–4; high 5–6; very high ≥ 7). Adjusted odds ratios (aOR) and their 95% confidence intervals (CI) were estimated using multivariable logistic regression models including age, BMI, family history of diabetes mellitus, previous GDM, miscarriages, and gravidity. Overall, middle-high MD adherence was 216/291 (74.2%) and very high adherence was 17/291 (5.8%) in cases. In controls the corresponding figures were 900/1175 (76.6%) and 73/1175 (6.2%), respectively. Compared to low adherence, high MD adherence was associated with GDM reduction (aOR 0.61, 95% CI 0.39,0.94; *p* = 0.028), and very high MD adherence was even more strongly associated (aOR 0.33, 95% CI 0.15, 0.72; *p* = 0.005). The protective effect of adherence to the MD prior to pregnancy should be considered as a preventive tool against the development of GDM.

## 1. Introduction

Gestational diabetes mellitus (GDM), a state of carbohydrate intolerance which develops or is first recognized in the second or third trimester of pregnancy [1], is an important public health problem that affects both mother and offspring. The complications of GDM include spontaneous abortion, fetal anomalies, preeclampsia, stillbirth, macrosomia, hypoglycaemia, and neonatal hyperbilirubin, amongst others [1,2]. It is estimated that the prevalence of GDM has been increasing worldwide, growing in parallel with obesity [1]. Its prevalence ranges between 2.5% and 14% influenced by racial, geographic and dietary factors [3,4], reaching almost 20% in some Asian countries [5].

The maternal diet composition affects the metabolic patterns of both mother and offspring [6,7,8,9,10]. The Mediterranean diet (MD) is associated with improved health outcomes [11], with a greater adherence to a MD pattern linked to lower cardiovascular disease [12,13] and risk factors (i.e., reduced obesity [14], hypertension [15]), the prevention of some cancers (i.e., breast, endometrium, ovary, prostate, and stomach [16]) and reduced incidence of micronutrient deficiencies [17]. There is also current scientific evidence regarding the protective effects of the MD pattern on type-2 diabetes [18]. As GDM shares the physiopathological mechanisms of diabetes mellitus, a MD may act as a protective factor for its development. Studies of dietary advice including a MD during pregnancy [7,19,20,21,22,23,24], suggest a posible benefit.

Previous studies have analyzed the association between adherence to a MD, or other dietary compositions during pregnancy, and GDM development [19,20,21,22,23,24], yet few have assessed the relationship between pre-pregnancy adherence to a MD and the development of GDM. Those that do have inconsistent results and show a less clear association, which may be due to the fact that some of these studies have been carried out on the MD in non-Mediterranean populations [25,26,27] or with different anthropometric, sociodemographic characteristics and culinary habits [19,20,21,22,23,24,25,26,27,28]. Thus, this association has not yet been demonstrated consistently or conclusively. We evaluated the effect of the pre-pregnancy Mediterranean diet (MD) level of exposure on the odds of GDM development.

## 2. Material and Methods

### 2.1. Study Design and Setting

This study was a case-control study consisting of pregnant women with GDM (cases) and those without (controls) in the catchment area of Virgen de las Nieves University Hospital of Granada, Spain (Project of Excellence of the Junta de Andalucía CTS 05/942). Ethical approval was obtained through the Ethics and Research Committees of the University of Granada and the Virgen de las Nieves University Hospital of Granada. One in five women who attended the antenatal visit for the screening ultrasound scan at 20–22 week of gestation were systematically informed about the study and informed consent was obtained for participation. The antenatal protocol in the South of Spain includes a systematic visit to the obstetrician at 20 weeks for all pregnant women [29]. Sample size was calculated considering all the following assumptions: case to control ratio 1:4, percentage of controls exposed to a moderate/high adherence to a MD pattern of 50% [30], an odds ratio (OR) greater than or equal to 1.5 for a population that does not have a MD, accepting an alpha risk of 0.05 and a beta risk of 0.2 in a two-sided test. The sample size was calculated using Fleiss’s formula with correction of continuity, and a total of 255 cases and 1020 controls was the estimation [31].

### 2.2. Participants

The participants consisted of Spanish women over 18 years of age with a low risk pregnancy. Women with a diagnosis of type 1 or 2 diabetes, or carbohydrate intolerance prior to pregnancy, as well as high risk pregnancies and those that needed to modify their diet or physical activity level in the previous year or during the first half of gestation for a medical reason were excluded.

Following the universal 50 g glucose challenge test in gestational weeks 24–28, women who had a venous plasma glucose ≥140 mg/dL were scheduled for a diagnostic 3 h, 100 g, oral glucose tolerance test. The National Diabetes Data Group (NDDG) criteria (fasting, 105 mg/dL; 1 h, 190 mg/dL; 2 h, 165 mg/dL; 3 h, 145 mg/dL) were considered [32]. GDM (cases) was defined as at least two plasma glucose measurements equal to or higher than the cutoff points. The control group had a negative 50 g glucose challenge test (<140 mg/dL) or positive 50 g glucose challenge test (≥140 mg/dL) and negative diagnostic oral glucose tolerance test.

### 2.3. Data Sources and Variables

Information about the pregnant women was collected on: anthropometric data, sociodemographic variables and personal, obstetric and family history, as well as her current work situation.

#### 2.3.1. Dietary Assessment

To collect information on the dietary pattern of the women, the food consumption frequency questionnaire (FFQ) developed by Martín-Moreno et al. was used. This questionnaire has been translated, adapted, and validated in the Spanish population [33] and records the intake of 118 different foods. We collected the frequency of consumption and average amount for different food groups during the year prior to pregnancy. The interviews were always carried out prior to the visit to the obstetrician and by personnel trained for that purpose, with an approximate duration of 45 min.

To measure the adherence to the MD, the index developed by Trichopoulou et al. [34] was used. This index considers the following nine components: vegetables, legumes, fruits and nuts, cereals, fish, meat, dairy products, the ratio of monounsaturated lipids to saturated lipids and ethanol consumption. The median for each food group was estimated using the control group. For consumption of each typical Mediterranean food higher than the median of the consumption distribution in the control group, a person received 1 point; consumption lower received zero points. For consumption of non-Mediterranean foods lower than the median 1 point was awarded; consumption higher than the median received zero points. For ethanol consumption, only the intake of wine was taken into account, if it was between 5 and 25 g/day, women received 1 point and 0 if the value was higher or lower than that figure. The total score ranged from 0 (minimum adherence to a traditional MD pattern) to 9 (maximum adherence). Subsequently, this MD adherence variable was categorized as: low adherence (0–2), middle (3–4), high (5–6) and very high (≥7 points).

#### 2.3.2. Other Variables

Current smokers were defined as those who smoked at least one cigarette per day in the last six months. The educational level of women was registered as: primary studies (eight years or less of basic education); secondary (four years of secondary education) and university (university or postgraduate studies). Body mass index (BMI) was calculated as weight (kg) divided by height (m) squared. Both, weight and height just before pregnancy, were obtained from the woman’s medical records where it had been recorded by their doctor or nurse. The cutoff points of the World Health Organization were used to determine overweight and obesity in the participants. Women with a BMI ≥ 30 kg/m^2^ were classified as obese and those with a BMI ≥ 25 kg/m^2^ but <30 kg/m^2^ as overweight [35].

### 2.4. Statistical Analysis

In the descriptive analysis of the sample, the mean, standard deviation (SD) and range of the continuous quantitative variables were calculated: age, previous BMI, energy intake. Food intakes were adjusted for total energy intake using the residuals method for cases and controls as recommended by Willet et al. [36]. For the qualitative variables of interest, the distribution of absolute and relative frequencies was calculated. We identified the relationship between each of the components of the MD and the development of GDM using multivariable logistic regression models and calculated crude (cOR) and adjusted odds ratios (aOR), and their 95% confidence intervals (CI). We used information from previous studies and directed acyclic graph (DAG) to identify potential confounding, thus epidemiological and statistical criteria were used to construct the models. Age, BMI, family history diabetes mellitus, previous GDM, previous miscarriages, gravidity, total energy intake, and leisure time physical activity were taken into account as possible confounding factors. The statistical program Stata v.14 (Stata Corp., 2015, College Station, TX, USA) was used.

## 3. Results

There were 299 cases of pregnant women diagnosed with GDM and 1,222 controls without. Among the cases, one (0.3%) did not have the correct tests and seven (2.4%) decided not to participate after recruitment. Thus, eight cases (2.7%) were excluded from analysis. Among the controls, 13 (1.1%) did not participate, 19 (1.5%) did not complete the interview and 15 (1.2%) had data missing for other variables. Therefore, the final sample analysed included 291 cases and 1175 controls (Figure 1).

The age range of the participants was between 18–45 years (Table 1). The average age in cases was higher than in controls (33.50 years (SD 5.5) vs. 29.80 years (SD 5.1)). A greater frequency of antecedents of diabetes mellitus and previous GDM was observed among the cases than controls. The BMI was higher among cases than controls: 27.62 kg/m^2^ (SD 6.2) vs. 24.22 kg/m^2^ (SD 4.5), respectively. The cases had more frequent extreme scores on the global index (Figure 2). Middle-high adherence to the MD was 216/291 (74.2%) in cases and 900/1175 (76.6%) in controls (very high adherence of 5.8% vs. 6.2%, respectively).

Table 2 shows the average consumption of the components of the diet in each group of participants, with the average consumption of legumes in both being very similar. When the relationship between the consumption of each component of the MD pattern and the development of GDM was analyzed (Table 3), a statistically significant association was found only between the consumption of meat products and their derivatives and the development of GDM (aOR = 0.56; 95% CI 0.42, 0.74).

Middle-high adherence was very similar in both cases (74.2%) and controls (76.6%), with only a very high adherence of 5.8% vs. 6.2%, respectively (Table 4). In the crude analysis, the level of adherence to the MD was observed to increase the protective effect on the development of GDM. In the adjusted analysis, it was found that the strength of the association became more intense, so the aOR was increasingly protective as the level of adherence to the MD increased. The aOR = 0.61 (95% CI 0.39, 0.94); *p* = 0.028 and aOR = 0.33 (95% CI 0.15, 0.72), *p* = 0.005 for a high and very high adherence to the MD, respectively.

## 4. Discussion

Our results show a protective effect of adherence to the MD prior to pregnancy for preventing GDM, with a temporal association. Very high adherence to the MD was more strongly associated with a reduction in GDM suggesting a dose-response. In addition, we observed the protective role of low consumption of meat and derivatives on the development of GDM.

The strengths of our study include the large representative sample from a reference population healthy pregnant women in the South of Spain. Only a small number of participants were lost to the antenatal care protocol. There was approximately a 99% coverage of the population of pregnant women in the public hospital. The analysis of overall dietary patterns offered a global assessment using a validated FFQ in the Spanish population [33]. This approach is superior to evaluating individual food groups [37,38,39]. To minimize selection bias, the sample was recruited through systematic sampling, using the antenatal ultrasound which is mandated as part of routine care. The collection of dietary information pre-pregnancy made it possible to study a temporal relationship.

The possible limitations of the study include concern about recall accuracy but this is likely to be non-differential between cases and controls as the participants were interviewed before being evaluated for GDM. There may also be concern about social desirability bias [40], depending on what women think they should consume, but this would be directed toward the null avoiding invalidation of our observed results. In observational epidemiologic studies, effect sizes can be caused by residual confounding due to the presence of unknown factors. In the present study this has been addressed by using multivariable analyses, however, it can not be completely ruled out when interpreting our results. Additionally, RCT evidence during pregnancy is consistent with our findings [41,42,43].

Other studies that evaluated adherence to the MD have used different methods. For example, Tobias et al. [25] used the Trichopoulou index, adapting it by not including dairy products. They studied women without GDM in previous pregnancies. Exclusion of women with previous diabetes mellitus may concentrate nulliparous women in the dataset. In our study there were patients with a history of previous GDM both among cases and controls, increasing the generalisability. Although pregnant women with previous GDM may modify their dietary patterns before another pregnancy, we took into account the dietary pattern during the year prior to pregnancy.

In the assessment of the quality of diet, other studies have separated the effects of different foods or meals. However, we eat nutrients through food, and dietary patterns rich in one nutrient tend to be associated with greater or lesser consumption of others [44]. The demonstrated benefits of a MD are probably not due to the isolated effect of some specific component of it, but it is due to synergistic effects and complex interactions between all the rations components. This is probably why when comparing each one of the MD components individually, no significant results are obtained, except when the consumption of meat is analyzed. This result is consistent with Schoenaker et al. [27], who state that the pattern `meats, sandwiches and sweets´ was associated with an increased risk of GDM after adjustment for socioeconomic, reproductive and lifestyle factors. Other studies corroborate the association between the consumption of meat products and an increase in the risk of development of diabetes mellitus [21,45].

## 5. Conclusions

The protective effect of adherence to a MD pattern prior to pregnancy should be considered as a preventive tool against the development of GDM. The MD should be promoted during the pre-pregnancy period for maternal and offspring health. Health care providers should keep this conclusion in mind to encourage adherence to the MD in women.

## Figures and Tables

**Figure 1 nutrients-11-01003-f001:**
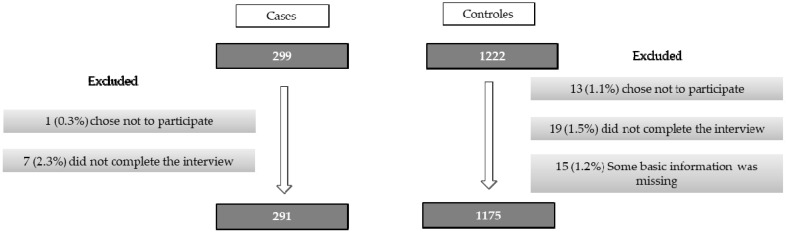
Flow diagram of the women included in the study and analyses.

**Figure 2 nutrients-11-01003-f002:**
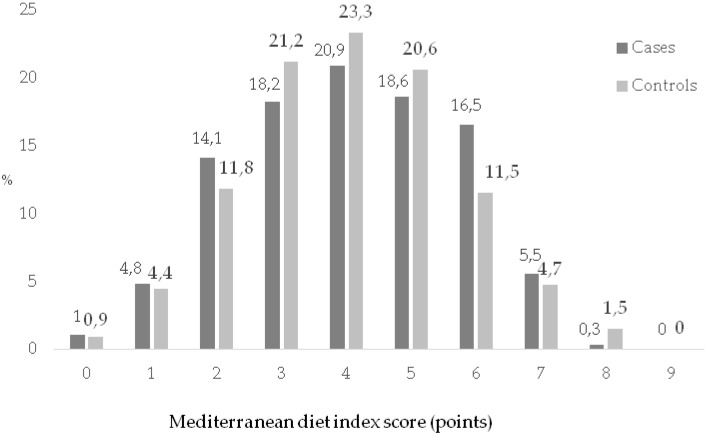
Distribution of adherence to Mediterranean diet in the study population.

**Table 1 nutrients-11-01003-t001:** Description of cases with gestational diabetes mellitus and the controls without gestational diabetes mellitus.

	Cases (*n* = 291)	Controls (*n* = 1175)	*p* Value
**Age (Mean; SD)**	33.50; SD = 5.5	29.80; SD = 5.1	<0.001
**(Years)**	*n* (%)	*n* (%)	
<25	18 (6.2)	178 (15.2)	
25–29	49 (16.8)	345 (29.4)	
30–34	91 (31.3)	436 (37.1)	
≥35	133 (45.7)	216 (18.3)	
**Education**			0.140
University	80 (27.5)	358 (30.5)	
Secondary	74 (25.4)	339 (28.8)	
Primary	137 (47.1)	478 (40.7)	
**Employment**			<0.001
Work outside the home	94 (32.4)	558 (47.5)	
Unemployment	24 (8.3)	84 (7.2)	
Sick leave in pregnancy	61 (21.0)	105 (8.9)	
Retired	2 (0.7)	6 (0.5)	
Housewife	109 (37.6)	421 (35.9)	
**Antecedents of Diabetes Mellitus**	135 (46.4)	300 (25.5)	<0.001
**Previous Gestational Diabetes Mellitus**	58 (19.9)	23 (1.9)	<0.001
**Gravidity**			<0.001
0	106 (36.4)	555 (47.2)	
1	89 (30.6)	365 (31.1)	
2	57 (19.6)	168 (14.3)	
3	22 (7.6)	61 (5.2)	
≥4	17 (5.8)	26 (2.2)	
**Parity**			<0.001
0	146 (50.2)	631 (53.7)	
1	85 (29.2)	416 (35.4)	
2	42 (14.4)	108 (9.2)	
≥3	18 (6.2)	20 (1.7)	
**Miscarriage**			<0.001
0	201 (69.1)	933 (79.4)	
1	69 (23.7)	199 (16.9)	
≥2	21 (7.2)	43 (3.7)	
**History of macrosomia**	10 (3.4)	37 (3.1)	0.062
**Body Mass Index (kg/m^2^)**			<0.001
(Mean; SD)	27.62; SD = 6.2	24.22; SD = 4.5	
18.5–24.9	117 (40.2)	789 (67.2)	
25–29.9	83 (28.5)	268 (22.8)	
≥30	91 (31.8)	118 (10.0)	
**Smoking**			0.161
Never	110 (37.8)	504 (42.9)	
Ex-smoker	73 (25.1)	242 (20.6)	
Current smoker	108 (37.1)	429 (36.5)	

**Table 2 nutrients-11-01003-t002:** Consumption of the components of the Mediterranean diet in the cases with gestational diabetes mellitus and the controls without gestational diabetes mellitus.

Components of the MD	Cases (*n* = 291) Mean (SD) 95% CI p25, p50, p75	Controls (*n* = 1175) Mean (SD) 95% CI p25, p50, p75	*p* Value
Vegetables (g/day)	584.14 (294.03)	588.72 (314.88)	0.082
	550.22–618.07	570.69–606.74	
	355.95, 560.71, 738.09	345.24, 540.48, 795.24	
Fruits (g/day)	241.34 (185.86)	217.86 (150.69)	0.023
	219.89–262.78	209.24–226.49	
	125.16, 207.44, 313.86	111.72, 191.30, 291.36	
Legumes (g/day)	0.23 (0.12)	0.23 (0.13)	0.954
	0.22–0.24	0.22–0.24	
	0.17, 0.22, 0.27	0.17, 0.22, 0.27	
Cereals (g/day)	236.47 (98.84)	227.90 (89.02)	0.151
	225.07–247.88	222.80–232.99	
	173.50, 227.14, 287.14	162.86, 227,14, 278.57	
Fish (g/day)	89.86 (61.59)	80.75 (50.18)	0.008
	82.76–96.97	77.88–83.63	
	47.26, 74.19, 121.43	47.26, 70.71, 107.62	
Dairy products (g/day)	474.31 (287.95)	492 (283.38)	0.342
	441.09–507.53	475.78–508.22	
	275, 397.62, 632.26	286.90, 439.52, 648.80	
Meat and derivatives (g/day)	172.92 (76.02)	149.91 (70.65)	<0.001
	164.15–181.69	145.87–153.95	
	119.28, 163.45, 208.69	103.09, 141.07, 184.28	
Ratio of monounsaturated/saturated lipids	0.98 (0.18)	0.92 (0.14)	0.695
	0.95–1.00	0.92–0.93	
	0.85, 0.94, 1.06	0.83, 0.91, 1.00	
Ethanol (g/day)	0.55 (1.37)	0.60 (1.83)	<0.001
	0.39–0.71	0.49–0.70	
	0, 0, 0.66	0, 0, 0.33	

g/day: grams/day. p: percentile.

**Table 3 nutrients-11-01003-t003:** Relationship between the components of the Mediterranean diet and the development of gestational diabetes mellitus.

Components of the MD	cOR	(95 % CI)	aOR	(95% CI)	*p* Value
**Vegetables**					
≥Median	1	Reference	1	Reference	
<Median	1.15	(0.89 , 1.49)	0.95	(0.69 , 1.29)	0.753
**Fruits**					
≥Median	1	Reference	1	Reference	
<Median	1.17	(0.91, 1.52)	0.84	(0.62, 1.14)	0.282
**Legumes**					
≥Median	1	Reference	1	Reference	
<Median	0.87	(0.67, 1.13)	0.75	(0.55, 1.01)	0.066
**Cereals**					
≥Median	1	Reference	1	Reference	
<Median	1.00	(0.78, 1.30)	0.79	(0.58, 1.06)	0.125
**Fish**					
≥Median	1	Reference	1	Reference	
<Median	1.00	(0.78, 1.30)	0.81	(0.61, 1.08)	0.163
**Dairy products**					
≥Median	1	Reference	1	Reference	
<Median	1.28	(0.99, 1.66)	1.25	(0.95, 1.64)	0.104
**Meat and derivatives**					
≥Median	1	Reference	1	Reference	
<Median	0.53	(0.41, 0.70) *	0.56	(0.42, 0.74) *	0.000
**Ratio of monounsaturated/saturated lipids**					
≥Median	1	Reference	1	Reference	
<Median	1.35	(1.04, 1.74) *	1.13	(0.85, 1.51)	0.381
**Ethanol**					
<5 and >25 g/day	1	Reference	1	Reference	
5–25 g/day	0.67	(0.26, 1.73)	0.61	(0.21, 1.74)	0.361

cOR: crude odds ratio; aOR: adjusted odds ratio, adjusted for age, BMI, family history DM, previous GDM, miscarriages, gravidity, total energy intake, and leisure time physical activity. * Significant association *p* < 0.05. g/day: grams/day.

**Table 4 nutrients-11-01003-t004:** Relationship between adherence to the Mediterranean diet and gestational diabetes mellitus.

Adherence to the MD (level)	Cases (*n* = 291)	Controls (*n* = 1175)	cOR	95% CI	aOR	95% CI	*p* Value
	*n* (%)	*n* (%)					
Low (0–2)	58 (19.9)	202 (17.2)	1	Reference	1	Reference	Reference
Middle (3–4)	114 (39.1)	523 (44.5)	0.75	(0.53, 1.08)	0.67	(0.44, 1.01)	0.060
High (5–6)	102 (35.1)	377 (32.1)	0.94	(0.65, 1.35)	0.61	(0.39, 0.94) *	0.028
Very high (≥7)	17 (5.8)	73 (6.2)	0.81	(0.44, 1.48)	0.33	(0.15, 0.72) *	0.005
*p* trend							0.014

cOR: crude odds ratio; aOR: adjusted odds ratio, adjusted for age, BMI, family history DM, previous GDM, miscarriages, gravidity, total energy intake, and leisure time physical activity. GDM: Gestational Diabetes Mellitus; * Significant association *p* < 0.05.

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
