# Peer review of "Pre-Pregnancy Adherence to the Mediterranean Diet and Gestational Diabetes Mellitus: A Case-Control Study"

_nutrients, 2019, doi:10.3390/nu11051003_

Reviewer 1 Report

The major limitation of this study is its cross-sectional nature.

The two groups (cases and controls) are quite different for many clinical characteristics.

I suggest to adjust analyses for education level too.

Author Response

Response to Reviewer 1

Comments and Suggestions for Authors

The major limitation of this study is its cross-sectional nature.The two groups (cases and controls) are quite different for many clinical characteristics.

Response:This work was a case-control study consisting of pregnant women with GDM (cases) and those without (controls).A case-control study is defined as an analytical study, which means that it is useful to approach a causality relationship (Rothman, 2012). Strictly speaking, we work with incidence cases of gestational diabetes and healthy pregnant, and information about diet refers to the pre-pregnancy period. Therefore, both the effect and the variable of exposure are not referring to the same moment of the time, and therefore it does not seem correct to define it as a cross-sectional study. According to Bradford-Hill causality criteria, the exposure must always be prior to the effect, and we think that our study design permits us to establish a correct temporal sequence. For this reason, we have not include this aspect among study limitations.

Regarding to the comment that “The two groups (cases and controls) are quite different for many clinical characteristics”, we agree with the reviewer. Nevertherless, the selection of the control group in a case-control study must be independent of both the level of exposure and the effect (Gestational Diabetes Mellitus). Therefore, it is logical that the frequency of potential risk factors related to the risk of Gestational Diabetes (age, previous gestational diabetes, overweight and obesity…) is higher in the group of cases than in the control group. Some of these variables are confounding variables and they are adjusted for in the analyses.

Reference:

-  Rothman KJ. Epidemiology: An Introduction. Second Edition,Oxford University Press, USA, 2012.

I suggest to adjust analyses for education level too.

Response:Thank you very much for this suggestion. However, education level did not behave as a confounding factor in our analyses. You can see the results of the analyses in the following tables, while the first one does not include the education level variable in the model, the second one does include it. This second model is not better than the previous one (Likelihood-ratio test = 0.38; p=0.8259. As can be observedthe OR for Mediterranean diet adherence does not change for both models –t_aj4 variable).

Table 1 results:

Logistic regression                             Number of obs     =      1,466

                                                LR chi2(22)       =     334.32

                                                Prob > chi2       =     0.0000

Log likelihood = -563.36875                     Pseudo R2         =     0.2288

------------------------------------------------------------------------------

caso | Odds Ratio   Std. Err.      z    P>|z|     [95% Conf. Interval]

-------------+----------------------------------------------------------------

t_aj4 |

        3-4  |   .6746591   .1409249    -1.88   0.060     .4480049    1.015982

        5-6  |   .6139779   .1360251    -2.20   0.028     .3977144     .947838

>6  |    .335421   .1312019    -2.79   0.005     .1558252    .7220094

             |

   edad_cat5 |

      25-29  |   1.542537   .4768797     1.40   0.161     .8415588    2.827398

      30-34  |   2.525481   .7610469     3.07   0.002     1.399061    4.558809

      35-39  |   6.195601   1.963269     5.76   0.000     3.329306    11.52957

>=40  |   37.01048   16.34903     8.17   0.000     15.57093    87.97006

             |

     imcrec4 |

    25-26,9  |   1.883462   .4380714     2.72   0.006     1.193926    2.971232

    27-29,9  |   2.050424   .4757967     3.09   0.002     1.301141    3.231196

>=30  |   4.893495   .9672872     8.03   0.000     3.321726     7.20899

             |

dmfamily |   2.060717   .3228051     4.62   0.000     1.515934     2.80128

dgpre |   10.94477   3.393426     7.72   0.000     5.960619    20.09656

             |

     abortos |

1  |   2.059423   .4837578     3.08   0.002     1.299567    3.263567

>=2  |   2.133861   1.073032     1.51   0.132     .7964008    5.717428

             |

   embarazos |

1  |   .5968298   .1202699    -2.56   0.010     .4020878    .8858906

2  |   .3954534    .111317    -3.30   0.001      .227766    .6865968

3  |   .2303767    .097959    -3.45   0.001     .1001146    .5301268

>=4  |    .437241   .2449749    -1.48   0.140     .1458191    1.311074

             |

 aflibrepre4 |

2  |   .7738127   .1651582    -1.20   0.230     .5092842    1.175741

3  |   .6362414   .1338832    -2.15   0.032     .4212162    .9610341

4  |   .6255011   .1344135    -2.18   0.029     .4105008    .9531081

             |

energiat |   1.000325   .0001068     3.04   0.002     1.000115    1.000534

       _cons |   .0372605   .0164173    -7.47   0.000      .015711    .0883681

------------------------------------------------------------------------------

Note: _cons estimates baseline odds.

Table 2 results

Logistic regression                             Number of obs     =      1,466

                                                LR chi2(24)       =     334.70

                                                Prob > chi2       =     0.0000

Log likelihood = -563.17743                     Pseudo R2         =     0.2291

------------------------------------------------------------------------------

caso | Odds Ratio   Std. Err.      z    P>|z|     [95% Conf. Interval]

-------------+----------------------------------------------------------------

       t_aj4 |

        3-4  |    .676117   .1413326    -1.87   0.061     .4488387    1.018482

        5-6  |   .6146511   .1362656    -2.20   0.028     .3980346    .9491535

>6  |   .3323963   .1302747    -2.81   0.005     .1541872    .7165789

             |

   edad_cat5 |

      25-29  |   1.588052   .4978967     1.48   0.140     .8589946    2.935886

      30-34  |   2.644065   .8213616     3.13   0.002     1.438308    4.860628

      35-39  |   6.519961   2.134514     5.73   0.000     3.432239    12.38547

>=40  |   39.26045   17.80458     8.09   0.000     16.14121    95.49362

             |

     imcrec4 |

    25-26,9  |   1.856768    .434195     2.65   0.008      1.17411    2.936341

    27-29,9  |   2.028357   .4721121     3.04   0.002     1.285352    3.200859

>=30  |   4.795055   .9619134     7.81   0.000      3.23622    7.104755

             |

dmfamily |    2.04976   .3215925     4.57   0.000     1.507147    2.787728

dgpre |   10.86007   3.375885     7.67   0.000     5.905188    19.97244

             |

     abortos |

1  |   2.082028   .4910055     3.11   0.002     1.311437    3.305413

>=2  |   2.194193   1.110014     1.55   0.120     .8140718    5.914076

             |

   embarazos |

1  |   .5865001   .1195775    -2.62   0.009     .3932987    .8746086

2  |   .3845013   .1097602    -3.35   0.001     .2197418    .6727952

3  |   .2215846   .0955902    -3.49   0.000     .0951342    .5161102

>=4  |    .408409   .2336537    -1.57   0.118     .1330812    1.253354

             |

 aflibrepre4 |

2  |   .7778883   .1662627    -1.18   0.240      .511664    1.182632

3  |   .6456439   .1367249    -2.07   0.039     .4263224    .9777952

4  |   .6351606   .1374589    -2.10   0.036     .4155947    .9707271

             |

energiat |   1.000316   .0001082     2.92   0.004     1.000104    1.000528

             |

    estudios |

Secundarios  |   1.072111    .222313     0.34   0.737     .7140593      1.6097

Primarios  |   1.134157   .2309664     0.62   0.536     .7609031    1.690506

             |

       _cons |   .0344842   .0159045    -7.30   0.000     .0139649    .0851535

------------------------------------------------------------------------------

Note: _cons estimates baseline odds.

Likelihood-ratio test.

LR chi2(2)= 0.38; Prob>chi2 = 0.8259.

Reviewer 2 Report

The manuscript “Pre-pregnancy adherence to mediterranean diet and gestational diabetes mellitus: A case-control study” is an interesting study but requires major revisions.

Firstly, the manuscript needs to be extensively proofread. There are multiple spelling mistakes, incorrect use of words, and incorrect sentences.

Please find below a list of more specific issues that need to be addressed throughout the manuscript:

1)      Mediterranean diet exposure is referred to as just ‘exposure’ but the authors report on levels of exposure, not binary yes or no. Therefore, throughout the manuscript it needs to be referred to as level of exposure.

2)      Why were middle and high combined together and not high-very high levels of exposure? It seems more logical to combine high-very high. Please change and reanalyse or justify in the manuscript why middle-high are grouped together.

3)      In line 40 – change ‘amongst other’ to’ amongst others’

4)      In line 40 – remove ‘the’ before ‘the GDM’

5)      In line 41 - prevalence of GDM is higher than 14% in some Asian countries, this sentence needs to be corrected and the appropriate figures reported

6)      Line 49 - need to expand on the relationship between MD and diabetes, include what the studies have found and the direction of the relationship. Far too vague as it currently stands.

7)      Also, there have been studies that have looked at pre-pregnancy Mediterranean diet and GDM risk – these need to be mentioned, their outcomes mentioned, and how the proposed study is novel/different to what has already been done in this area. There are also studies in Mediterranean populations so the detail in line 57 needs to be corrected. Examples of studies with relevant findings that should be discussed include:

Donazar-Ezcurra, M., Lopez-del Burgo, C., Martinez-Gonzalez, M.A., Basterra-Gortari, F.J., de Irala, J. and Bes-Rastrollo, M., 2017. Pre-pregnancy adherences to empirically derived dietary patterns and gestational diabetes risk in a Mediterranean cohort: the Seguimiento Universidad de Navarra (SUN) project. British Journal of Nutrition, 118(9), pp.715-721.

Tobias, D.K., Zhang, C., Chavarro, J., Bowers, K., Rich-Edwards, J., Rosner, B., Mozaffarian, D. and Hu, F.B., 2012. Prepregnancy adherence to dietary patterns and lower risk of gestational diabetes mellitus. The American journal of clinical nutrition, 96(2), pp.289-295.

8)      Line 81 - the NDGG criteria are not correctly described. This might be partly due to incorrect English language used. Please review and re-write.  

9)      Line 97 – don't talk about quality of diet – the study is measuring adherence to Mediterranean diet, not diet quality. Saying this implies that the MD is a high quality diet but the purpose of this study is to find that out, so that assumption should not be made.

10)   In regards to Mediterranean diet adherence, is all ethanol consumption grouped together? Please state this clearly in manuscript and the justification for this as one would think that red wine has different health effects to beer or whiskey, for example. 

11)   Remove ‘optimal consumption’ in line 105 or justify the use of the term ‘optimal’ if this is based on evidence.

12)   Line 108 to 111 – It is not clear what is meant by this statement

13)   More detail is required on screening for GDM that occurred prior to an OGTT – did ALL women undergo an initial screening test and if they failed then went on to OGTT and if they went on to an OGTT but weren’t diagnosed on OGTT then they weren’t considered for case or control group? This needs to be clearer in manuscript.

14)   Line 109 – check spelling of assessment

15)   Line 117 – at what pre-pregnancy timepoint were weight and height collected from medical records? These could vary hugely between participants and this needs to be stated as a limitation if they weren’t all collected at a similar time pre-pregnancy.

16)   Energy intake should be included as a confounding variable in the models. The models need to be re-analysed with energy intake as a confounding variable.

17)   Line 130 – gravity should be gravidity 

18)   Need more detail on line 131  about what purpose stata was used for

19)   Line 134 – include percentages for these figures too, as per line 135 and 136 

20)    ‘Finally’ is used inappropriately in the Figure 1 title

21)   Please state somewhere in the manuscript what the criteria were for the different employment categories. It is not clear what the difference is between housewife and unemployed

22)   Include P-values on table 1 for each of the descriptive variables

23)   Table 2 title – remove an ‘in’ 

24)   Table 2 – check spelling of Meats

25)   Include P-values in Table 2 for each component comparison

26)   Correct ‘derivatives’ in Table 3 (labelled currently as derivates) 

27)   Why are food groups split into those groups in Table 3? Why have they not been analysed as grams in a linear regression, as per grams displayed in table 2? If this was to be remedied then combining table 2 and table 3 could be sensible

28)   Also, why are medians in table 3 but means in table 2? Is the data normally distributed? If not, then medians and upper and lower quartiles should be reported in table 2.

29)   The formatting in table 4 makes it difficult to follow. The bolding and underline of part of the second row should be un-bolded and line removed

30)   P-values in table 4 should have ‘.’ not ‘,’ 

31)   It’s not clear what the reference population is that is referred to in the discussion but if the authors are referring to the hospital population, I don’t think this is appropriate. The reference population should be Spanish pregnant women if that is the population these findings would be translated to, not just from one hospital.

32)   Remove sentence in Line 181, as per comment above. Also, the sample was 1 in every 4 so the population was not covered by 99%. 

33)   In line 183 – this study is not an intervention so do not refer to it as an intervention

34)   Correct sentence in line 184 – “to evaluating individual food groups” 

35)   In line 185 – it is unclear why ultrasound was important for this study/outcome of interest. Please explain relevance in manuscript or remove.

36)   Need more emphasis in introduction and discussion as to what this study adds, why is it novel and necessary. Not clear what this study adds, particularly as stated in discussion that RCTs have already confirmed this same result, and observational studies have already been conducted.

37)   Need to acknowledge in limitations that the maternal diet might play a role also in GDM development, as may gestational weight gain and both could be significant confounders. 

38)   In line 212 – the last sentence is irrelevant, please remove.

39)   Remove last sentence of conclusion as that is beyond the scope of this study, MD during and after pregnancy were not addressed in this study to draw those conclusions.

Author Response

Response to Reviewer 2

Comments and Suggestions for Authors

The manuscript “Pre-pregnancy adherence to mediterranean diet and gestational diabetes mellitus: A case-control study” is an interesting study but requires major revisions.Firstly, the manuscript needs to be extensively proofread. There are multiple spelling mistakes, incorrect use of words, and incorrect sentences.

Response:Thanks for your suggestion and we apologise for the spelling mistakes, incorrect use of words, and incorrect sentences. One of the authors is English and member of the School of Medicine of Queen Mary University London. Moreover, we have sent this last version –after including reviewrs’ suggestions– to a native English reviewer and we also attach a certificate of this revision. That document certifies that this manuscript was edited for English language, grammar, punctuation, spelling, and style by Dr. Ingrid de Ruiter, MBChB, PhD, native English speaker and medical writer and editor at medicalwriting.es.

Please find below a list of more specific issues that need to be addressed throughout the manuscript:

1) Mediterranean diet exposure is referred to as just ‘exposure’ but the authors report on levels of exposure, not binary yes or no. Therefore, throughout the manuscript it needs to be referred to as level of exposure.

Response:Thank you. We agree with you. We have modified the word “exposure” by “level of exposure” in the objective of the manuscript(lines 20 and 63in Abstract and Introduction sections respectively).

2) Why were middle and high combined together and not high-very high levels of exposure? It seems more logical to combine high-very high. Please change and reanalyse or justify in the manuscript why middle-high are grouped together.

Response:The groups of medium and high adherence to the Mediterranean dietwere not combined, except in the Abstract and Results sections (lines 26-28 and 195-196) to describe the characteristics of cases and controls. The analysis of the relationship between adherence to the Mediterranean diet (low, middle, high and very high) and development of GDM, is shownfor each category in table 4 independently. Moreover, the distribution of adherence to Mediterranean diet is presented in Figure 2 without specifying concrete categories.

3)  In line 40 – change ‘amongst other’ to’ amongst others’

Response: Thanks for your suggestion. We have modified the sentence according to your suggestions (line 40).

4)  In line 40 – remove ‘the’ before ‘the GDM’

Response: Thanks for your suggestion. The word “the” has been removed (line 40).

5)  In line 41 - prevalence of GDM is higher than 14% in some Asian countries, this sentence needs to be corrected and the appropriate figures reported.

Response: The sentence "reaching almost 20% in some Asian countries" has been added and a new reference has been included (reference 5: Zhu WWet al.,2017) (lines 42 and 43). Hencethe numbering of the references has changed in the text and in the References section comparing with the first version of the manuscript.

New reference:

-       Zhu WW, Yang HX, Wang C, Su RN, Feng H, KapurA.High Prevalence of Gestational Diabetes Mellitus in Beijing: Effect of Maternal Birth Weight and Other Risk Factors. Chin Med J (Engl). 2017;130:1019-1025.

6) Line 49 - need to expand on the relationship between MD and diabetes, include what the studies have found and the direction of the relationship. Far too vague as it currently stands.

Response: This sentence has been changedto: “There is also current scientific evidence of protective effects of the MD on type-2 diabetes” (lines 52-55). The reference is maintained

7) Also, there have been studies that have looked at pre-pregnancy Mediterranean diet and GDM risk – these need to be mentioned, their outcomes mentioned, and how the proposed study is novel/different to what has already been done in this area. There are also studies in Mediterranean populations so the detail in line 57 needs to be corrected. Examples of studies with relevant findings that should be discussed include:

-  Donazar-Ezcurra, M., Lopez-del Burgo, C., Martinez-Gonzalez, M.A., Basterra-Gortari, F.J., de Irala, J. and Bes-Rastrollo, M., 2017. Pre-pregnancy adherences to empirically derived dietary patterns and gestational diabetes risk in a Mediterranean cohort: the Seguimiento Universidad de Navarra (SUN) project. British Journal of Nutrition, 118(9), pp.715-721.

-  Tobias, D.K., Zhang, C., Chavarro, J., Bowers, K., Rich-Edwards, J., Rosner, B., Mozaffarian, D. and Hu, F.B., 2012. Prepregnancy adherence to dietary patterns and lower risk of gestational diabetes mellitus. The American journal of clinicalnutrition, 96(2), pp.289-295.

Response:We agree with your assessment and we have clarified this in the text(last paragraph of Introduction section – lines 56-63–). We want also to thank you for your reference to Tobias DK et al and Donazar-Ezcurra M et al cites, inasmuch as we have detected a mistake in the Tobias reference, and so it has been replaced in the new version of the manuscript. Regarding the conclusions of these two articles, we can see how the results are not yet consistent:

1)    Donazaret al., 2017: “No association was found between adherence to the Mediterranean dietary pattern (MDP) and GDM incidence (OR 1.08; 95% CI 0.68, 1.70 for the highest quartile compared with the lowest)”.

2)    Tobiaset al.,2012: “Prepregnancy adherence to healthful dietary patterns is significantly associated with a lower risk of GDM”.

As can be observed, the results are not still consistent, and we think it is necessary to expand the research in this area.

8)  Line 81 - the NDGG criteria are not correctly described. This might be partly due to incorrect English language used. Please review and re-write. 

Response: Thank you, we have rewritten that paragraph to improve your understanding. This sentence has been changed to: “Following the universal 50 g glucose challenge test in gestational weeks 24-28, women who had a venous plasma glucose ≥140 mg/dl were scheduled for a diagnostic 3-hour 100 g, oral glucose tolerance test. The National Diabetes Data Group (NDDG) criteria (fasting, 105 mg/dl; 1 h, 190 mg/dl; 2 h, 165 mg/dl; 3 h, 145 mg/dl) were considered [35]. GDM (cases) was defined as at least two plasma glucose measurements equal to or higher than the cutoff points. The control group had a negative 50 g glucose challenge test (<140 mg/dl) or positive 50 g glucose challenge test (≥140 mg/dl) and negative diagnostic oral glucose tolerance test.”.

New reference:

- National Diabetes Data Group. Classification and diagnosis of diabetes mellitus and other categories of glucose intolerance. Diabetes. 1979;28:1039–1057. The numbering of the references changes in the text and in the References section.

9)  Line 97 – don't talk about quality of diet – the study is measuring adherence to Mediterranean diet, not diet quality. Saying this implies that the MD is a high quality diet but the purpose of this study is to find that out, so that assumption should not be made.

Response: The sentence “To measure the quality of the diet, the index of adherence…” has been changed to: “To measure the adherence to MD, the index developed by Trichopoulou et al.…”(line 106).

10) In regards to Mediterranean diet adherence, is all ethanol consumption grouped together? Please state this clearly in manuscript and the justification for this as one would think that red wine has different health effects to beer or whiskey, for example.

Response:In this article, only the consumption of wine has been taken into account. We have now clarified this in the text, we have added "only the intake of wine was taken into account" (lines113-114). This decision was made according to the index developed by Trichopoulou et al (2003) “…a low intake of meat and poultry, and a regular but moderate intake of ethanol, primarily in the form of wine and generally during meals…”.

References:

-  Trichopoulou A, Costacou T, Bamia C & Trichopoulou D. (2003) Adherence to a Mediterranean and survival in a Greek population. N Engl J Med 348:2599–2608.

11) Remove ‘optimal consumption’ in line 105 or justify the use of the term ‘optimal’ if this is based on evidence.

Response: Thanks for your suggestion. We have removed "optimal comsumption" from the manuscript (lines 113-115). As previously commented, and according to Trichopoulou et al (2003): "For ethanol, a value of 1 was assigned to men who consumed between 10 and 50 g per day and to women who consumed between and 25 g per day", and the variable was defined according to Trichopoulou’s reference.

Reference:

- Trichopoulou, A.; Costacou, T.; Bamia, C.; Trichopoulos, D. Adherence to a Mediterranean diet and survival in a Greek population. N. Engl. J. Med. 2003, 348, 2599-2608.

In table 3, “optimal” and “suboptimal consumption” terms have been replaced by “5-25 g/d” and “<5 and="">25 g/d” respectively.

12)  Line 108 to 111 – It is not clear what is meant by this statement

Response: We agree with you and the sentence has been eliminated.

13)  More detail is required on screening for GDM that occurred prior to an OGTT – did ALL women undergo an initial screening test and if they failed then went on to OGTT and if they went on to an OGTT but weren’t diagnosed on OGTT then they weren’t considered for case or control group? This needs to be clearer in manuscript.

Response: Yes, we agree more detail was required and we have clarified this aspect in our response to question 8, above. Furthermore,the use of the ADA criteria to identify GDM would result in a 31.8% increase in prevalence compared to the NDDG criteria. However, as the contribution of these additionally diagnosed cases to the adverse results of the GDM is not substantial in our environment, a change in the diagnostic criteria has not been considered justified (Ricart W et al., 2000).

Reference:

-  Ricart W, Lopez J, Mozas J, Pericot A, Sancho MA, González N, et al. Potential impact of American Diabetes Association (2000) criteria for diagnosis of gestational diabetes mellitus in Spain. Diabetologia. 2005; 48:1135–1141.

14)  Line 109 – checkspelling of assessment

Response: Thanks for your suggestion. According to question 12, the sentence has been previously eliminated.

15)  Line 117 – at what pre-pregnancy timepoint were weight and height collected from medical records? These could vary hugely between participants and this needs to be stated as a limitation if they weren’t all collected at a similar time pre-pregnancy.

Response: We agree with your opinion. The sentence “…Both, weight and height, were obtained from the woman's medical records” has been changed to “…Both, weight and heightjust before pregnancy, were obtained from the woman's medical records”. It should be clear that the weight and height corresponded to the pregestational period(lines 123-124). Spain’s National Health Service has a free prenatal care programme for all pregnant women, and the regional Andalusian Public Health Authority—with associated clinical guidelines for pregnancy, delivery, and puerperal care—recommends all pregnant women have an ultrasound early in the second trimester. At the appointment for this ultrasound, the women were invited to participate in our study. However, the follow-up of these women starts around 6-8 weeks of gestation by midwives who document the weight in the first visit as well as record the pre-pregnancy weight (a self-reported weight).

16)   Energy intake should be included as a confounding variable in the models. The models need to be re-analysed with energy intake as a confounding variable.

Response: Thanks for your suggestion. We realized that we did not include the following information in the methodology section(lines 129-131):"Food intakes were adjusted for total energy intake using the residuals method for cases and controls as recommended by Willet et al." Moreover, energy intake was included in the models and it has been referenced as part of the legend of the tables 3 and 4. The results shown in the tables are adjusted by energy intake (variable = energiat) and can be seen in the following results:

Logistic regression                             Number of obs     =      1,466

                                                LR chi2(22)       =     334.32

                                                Prob > chi2       =     0.0000

Log likelihood = -563.36875                     Pseudo R2         =     0.2288

------------------------------------------------------------------------------

caso | Odds Ratio   Std. Err.      z    P>|z|     [95% Conf. Interval]

-------------+----------------------------------------------------------------

       t_aj4 |

        3-4  |   .6746591   .1409249    -1.88   0.060     .4480049    1.015982

        5-6  |   .6139779   .1360251    -2.20   0.028     .3977144     .947838

>6  |    .335421   .1312019    -2.79   0.005     .1558252    .7220094

             |

   edad_cat5 |

      25-29  |   1.542537   .4768797     1.40   0.161     .8415588    2.827398

      30-34  |   2.525481   .7610469     3.07   0.002     1.399061    4.558809

      35-39  |   6.195601   1.963269     5.76   0.000     3.329306    11.52957

>=40  |   37.01048   16.34903     8.17   0.000     15.57093    87.97006

             |

     imcrec4 |

    25-26,9  |   1.883462   .4380714     2.72   0.006     1.193926    2.971232

    27-29,9  |   2.050424   .4757967     3.09   0.002     1.301141    3.231196

>=30  |   4.893495   .9672872     8.03   0.000     3.321726     7.20899

             |

dmfamily |   2.060717   .3228051     4.62   0.000     1.515934     2.80128

dgpre |   10.94477   3.393426     7.72   0.000     5.960619    20.09656

             |

     abortos |

1  |   2.059423   .4837578     3.08   0.002     1.299567    3.263567

>=2  |   2.133861   1.073032     1.51   0.132     .7964008    5.717428

             |

   embarazos |

1  |   .5968298   .1202699    -2.56   0.010     .4020878    .8858906

2  |   .3954534    .111317    -3.30   0.001      .227766    .6865968

3  |   .2303767    .097959    -3.45   0.001     .1001146    .5301268

>=4  |    .437241   .2449749    -1.48   0.140     .1458191    1.311074

             |

 aflibrepre4 |

2  |   .7738127   .1651582    -1.20   0.230     .5092842    1.175741

3  |   .6362414   .1338832    -2.15   0.032     .4212162    .9610341

4  |   .6255011   .1344135    -2.18   0.029     .4105008    .9531081

             |

energiat |   1.000325   .0001068     3.04   0.002     1.000115    1.000534

       _cons |   .0372605   .0164173    -7.47   0.000      .015711    .0883681

------------------------------------------------------------------------------

Note: _cons estimates baseline odds.

Reference:

-       Willett W, Stampfer M (1998) Implications of total energy intake for epidemiologic analyses. In Nutritional Epidemiology, 2nd ed.;Willett W., Ed.; Oxford University Press: New York, NY, USA.

17)   Line 130 – gravity should be gravidity

Response: Thanks for your appreciation. This change has been made on the lines 137-138.

18)   Need more detail on line 131  about what purpose stata was used for

Response: The statistical programme Stata v.14 (Stata Corp., LP, College Station, TX, USA) was used for alldata analyses. Stata is a complete, integrated software package that provides all your data science needs—data manipulation, visualization, statistics, and reproducible reporting—. The department has purchased the license for this software, so it is available for our use.

19) Line 134 – include percentages for these figures too, as per line 135 and 136

Response: We have includedthe percentages in the text: "Among the cases, one did not have the correct tests (0.3%) and 7 decided not to participate after recruitment (2.4%)" (line142-143).

20) ‘Finally’ is used inappropriately in the Figure 1 title

Response: Thanks for your suggestion. We have eliminated the word "finally" in the Figure 1 title.

21)Please state somewhere in the manuscript what the criteria were for the different employment categories. It is not clear what the difference is between housewife and unemployed

Response: The main difference is that unemployed, are those women who usually work for a company and has currently lost her job but they are looking for one.

The category housewife refers to those women who have decided to devote themselves to housework and do not plan to look for a job as an employee or become self-employed. They do not have economic remuneration for this activity.

To clarify the terms, the following classification has been used: 1. Work outside the home; 2. Unemployment; 3. Sick leave in pregnancy; 4. Retired; and 5. Housewife.

22)  Include P-values on table 1 for each of the descriptive variables

Response: P-values have been included in table 1.

23)  Table 2 title – remove an ‘in’ 9

Response: We have eliminated the word "in" in the Table 2 title.

24)  Table 2 – check spelling of Meats

Response: Thanks. The spelling of "meat" is reviewed in Table 2.

25)   Include P-values in Table 2 for each component comparison

Response: P-values have been included in Table 2. Moreover, the value of the median for each component of the index for Mediterranean diet has been included in Table 3.

26)   Correct ‘derivatives’ in Table 3 (labelled currently as derivates)

Response:The spelling of the term "derivatives" has been corrected in Table 3.

27)   Why are food groups split into those groups in Table 3? Why have they not been analysed as grams in a linear regression, as per grams displayed in table 2? If this was to be remedied then combining table 2 and table 3 could be sensible

Response: The main objective of our study is to analyze the relationship between a Mediterranean dietary pattern and gestational diabetes. For this reason, we have worked with a global index, the Trichopoulou index, to measure the level of adherence to a Mediterranean dietary pattern. We are not interested in analyzing the role of each component of the index separately, rather the joint role of the Mediterranean dietary pattern. Thererfore, the analyses conducted in the study and the information shown in the article are based on the Trichopoulou score and the way to construct it.

28)   Also, why are medians in table 3 but means in table 2? Is the data normally distributed? If not, then medians and upper and lower quartiles should be reported in table 2.

Response:The main measures of central dispersion in both tables are shown, but as we have commented in the previous question, the construction of the index assigns the score according to the median, not the mean. According to your suggestion, percentile 25, 50 and 75 have been included in table 2.

29)   The formatting in table 4 makes it difficult to follow. The bolding and underline of part of the second row should be un-bolded and line removed

Response: Thank you. We totally agree with you. It seems to be an edition mistake that we have already solved. It did not appear in the original version of the manuscript that was sentfor review.

30)   P-values in table 4 should have ‘.’ not ‘,’

Response: This editing error has beencorrected. We have changed ',' to'.'.

31)   It’s not clear what the reference population is that is referred to in the discussion but if the authors are referring to the hospital population, I don’t think this is appropriate. The reference population should be Spanish pregnant women if that is the population these findings would be translated to, not just from one hospital.

Response:Thanks for your suggestion. In the second paragraph of the discussion section we referred to the sampling population. We have completed the sentence to improve understanding. The phrase included in the manuscript in the current version is: “The strengths of our study include the large representative sample from a reference population of healthy pregnant women of the South of Spain”.

32)   Remove sentence in Line 181, as per comment above. Also, the sample was 1 in every 4 so the population was not covered by 99%.

Response: In Andalusia (Spain) the Maternal and Child Health Program of the Junta de Andalucía "Manual Pregnancy, Childbirth and Puerperio. Recommendations for Mothers and Fathers "was developed in the 80s and since that year, all pregnant women are recommended to havean ultrasound scanin week 20 of pregnancy (Manual Pregnancy, Childbirth and Puerperio, 2006). The coverage of this program is around 99% of the pregnant women in Granada, Spain (although it is true that there are around 5-10% of pregnant women that combine public and private cares for their pregnancies). Therefore, that sentence in the manuscript does not refer to the selection of the study sample.

Adifferent aspect is the selection of the sample. For the control group, we invited 1 in 5 of the eligible populationto participate to. They were invited when theyattending the second visit established in the Pregnancy Care Program.Asthe appointment for the performance of the ultrasound is systematic, without being conditioned by any type of specific pathology, we consider that this type of sampling did not introduce any type of selection bias.

33)   In line 183 – this study is not an intervention so do not refer to it as an intervention

Response: Thanks for your suggestion. We have removed that word from lines 220-221.

34)   Correct sentence in line 184 – “to evaluating individual food groups”

Response: The corrected sentence “to evaluating individual food groups” has been included (lines221-222).

35)   In line 185 – it is unclear why ultrasound was important for this study/outcome of interest. Please explain relevance in manuscript or remove.

Response: We have already commented on this in a previous question. There was approximately 99% coverage of the population of pregnant women in the public hospital, and all pregnant women routinelyhave an ultrasound scan in week 20 of pregnancy. Hence, this was a good time point in the pregnancy for recruitment of participants without introducing selection bias.

36)   Need more emphasis in introduction and discussion as to what this study adds, why is it novel and necessary. Not clear what this study adds, particularly as stated in discussion that RCTs have already confirmed this same result, and observational studies have already been conducted.

Response: Thanks for your question. We have modified the introduction section and pointed out the lack of consistency of previous studies, as we have also commented above in comment 7. The new paragraph is the following:”Few studies assess the relationship between pre-pregnancy adherence to MD and GDM. Some studies analyze the association between adherence to MD and GDM or adherence to other types of diets, evaluating the dietary pattern during, but not before pregnancy [21-26]. Those that do, have inconsistent results and with a less clear association, which may be due to the fact that some studies have been carried out on the MD in non-Mediterranean populations [27-29] or with different anthropometric, socio-demographic characteristics and culinary habits [21-30]. Thus, this association has not yet been demonstrated consistently or conclusively.”

On the other hand, the clinical trials that have been carried out to date have investigatedthe relationship between the adherence to the Mediterranean diet pattern during pregnancy and the risk of gestational diabetes, not pre-pregnancy diet (not before pregnancy). Of the three trials, one is based on a nutritional intervention in women with previous gestational diabetes and therefore does not have to correspond to a population like ours (Perrez-Ferreet al., 2015). The ESTEEM clinical trial trial has not yet published their results - March 19, 2019-. And the third trial by Assaf-Balut C et al.,(2015) does show results similar to ours, although it works with a very select population with an incidence of GDM of17.1% in the intervention group and 23.4% in the control group (incidence much higher than expected in the Spanish population). Therefore, we believe that our study adds moreevidence on the relationship between Mediterranean diet and gestational diabetes.

37)   Need to acknowledge in limitations that the maternal diet might play a role also in GDM development, as may gestational weight gain and both could be significant confounders.

Response: Thanks for the comment and suggestion. Most of the referenced studies that analyze the relationship between Mediterranean diet and gestational diabetes do not consider weight gain among the variables of interest (Gicevicet al., 2018; Donazar-Ezcurraet al., 2017; Schoenakeret al., 2015; Tobias et al., 2012). In addition,Chiefari E et al. (2017) in a current review on gestational diabetes, also does not specifically mention weight gain during pregnancy as a modifiable risk factor (Chiefari et al., 2017). Reasons why we do not consider this variable as a potential confoundingfactor.

However, after your comment, the relationship between maternal weight gain from the beginning of pregnancy to the time of the interview was analyzed, confirming a positive association with the risk of gestational diabetes but not with the level of adherence to a Mediterranean diet before pregnancy, neither for women with gestational diabetes nor for healthy women.

(Spearman's rho = -0.0457, Prob> |t| = 0.4399 para mujeres diagnosticadas de diabetes gestacional; Spearman's rho = 0.0001, Prob> |t| = 0.9962 para mujeres sanas).

-  Chiefari E, Arcidiacono B, Foti D, Brunetti A. Gestational diabetes mellitus: an updated overview. J Endocrinol Invest 2017; 40:899–909.

38)   In line 212 – the last sentence is irrelevant, please remove.

Response:This sentence has beeneliminatedalong withits corresponding reference (line 248).

39)   Remove last sentence of conclusion as that is beyond the scope of this study, MD during and after pregnancy were not addressed in this study to draw those conclusions.

Response:The conclusions have beenre-written as (302-304): “The protective effect of adherence to a MD pattern prior to pregnancy should be considered as a preventive tool against the development of GDM. A MD should be promoted during the pre-pregnancy period for maternal and offspring health. Health care providers should keep this conclusion in mind to encourage adherence to the MD in women”.

Reviewer 3 Report

The authors conducted a case-control study to evaluate the effect of pre-pregnancy Mediterranean diet (MD) exposure on the odds of Gestational Diabetes Mellitus (GDM).

Overall the study is well designed and well written.

Few points of concerns are:

1.    The numerous health benefits of the Mediterranean Diet (MedDiet) have been repeatedly reproduced. However, have you evaluated the effect of MD on gestational diabetes mellitus (GDM) complications (glycemic control and pregnancy outcomes), or comparing these women with those with normal glucose tolerance (NGT).

2.    Did you look at the HbA1c levels in these patients? It will be interesting to compare the glycemic control in these patients.

3.    How might these results change the focus of research or clinical practice?

4.    The diet of pregnant women give qualitative rather than quantitative information. Exact caloric and macronutrient intake is lacking because in GDM management women are not provided with a diet with a specific daily total caloric intake.

Author Response

Answer to Reviewer 3

Comments and Suggestions for Authors

The authors conducted a case-control study to evaluate the effect of pre-pregnancy Mediterranean diet (MD) exposure on the odds of Gestational Diabetes Mellitus (GDM). Overall the study is well designed and well written.

Response: Thank you very much for your comments on our manuscript (Manuscript ID nutrients-460745).

Few points of concerns are:

1.    The numerous health benefits of the Mediterranean Diet (MedDiet) have been repeatedly reproduced. However, have you evaluated the effect of MD on gestational diabetes mellitus (GDM) complications (glycemic control and pregnancy outcomes), or comparing these women with those with normal glucose tolerance (NGT).

Response:The potential effect on GDM complications would be an interesting study. However, the objective of this study was to assess the development of gestational diabetes mellitus (GDM) according to pregestational adherence to the Mediterranean diet (MD). We did not set ourselves the objective of assessing the effect of the MD on GDM complications in this case. On the other hand, this point would be more complicated, because it would need to integrate, from the dietary point of view, the continuity of adherence to the MD during pregnancy and the interference of the restrictive diet to which pregnant women with GDM are subjected. Moreover, the design of our study, a case control study, does not allow us to address the reviewer’s new proposed objectives.

2.    Did you look at the HbA1c levels in these patients? It will be interesting to compare the glycemic control in these patients.

Response: In the same sense of our previous reflection, we could not evaluate glycemic control or HbA1c levels in these women. It would be very interesting to do it, but it was outside our objectives and study design.

3.    How might these results change the focus of research or clinical practice?

Response: In the pregnancy the dietary recommendations that are given to the women are brief and oftenwas provided late. We should promote health in previous stages, for example to a young population of childbearing age. This could help to improve lifestyles and prevent different levels of disease, among others, gestational diabetes mellitus.

For that reason, we indicated in the conclusions of this study: “The protective effect of adherence to MD prior to pregnancy should be considered as a preventive tool against the development of GDM. A MD should be promoted during the pre-pregnancy period for maternal and offspring health. Health care providers should keep this conclusion in mind to encourage adherence to the MD in women”.

4.The diet of pregnant women give qualitative rather than quantitative information. Exact caloric and macronutrient intake is lacking because in GDM management women are not provided with a diet with a specific daily total caloric intake.

Response: The adherence to the Mediterranean diet as prevention of gestational diabetes mellitus is a qualitative approach. However, in the management of gestational diabetes mellitus, it is necessary that patients be recommended a diet with a total daily caloric intake and specific macronutrients according to body mass index.

We agree with your comment. In fact, it is according to last American Diabetes Association (ADA) recomendations published in Diabetes Care (Diabetes Care 2019 Jan; 42(Supplement 1): S165-S172). Regarding nutrition therapy for GDM, ADA says: “(…) Medical nutrition therapy for GDM is an individualized nutrition plan (…). The food plan should provide adequate calorie intake to promote fetal/neonatal and maternal health, achieve glycemic goals, and promote appropriate gestational weight gain. There is no definitive research that identifies a specific optimal calorie intake for women with GDM or suggests that their calorie needs are different from those of pregnant women without GDM.” Thus, we can not give an individualized recommendation in this regard, given the design of the study. Therefore, weagreethat more studies are needed.

Round  2

Reviewer 3 Report

The authors have pointed out some valid justifications to the reviewers questions. Additionally, Appropriate changes has been done in  the manuscript. Even though comparing HbA1c levels in the GDM patient adhering to the MD diet should be considered for future studies.

Author Response

In agreement. Thanks for your contribution.